# TrajTune: Trajectory-Based Prompt Optimization for Robust LLM Agents

## Abstract

Large Language Model (LLM) agents are increasingly deployed in complex tasks involving multi-step reasoning and dynamic API interactions. However, these agents often fail due to issues like hallucinated tool calls or repetitive actions, which are not effectively addressed by current prompt optimization methods that focus primarily on textual output quality.

We present TrajTune, a trajectory-aware prompt optimization framework designed to enhance the reliability and adaptability of LLM agents. TrajTune captures structured execution traces, computes fine-grained error metrics, and compares them against adaptive thresholds. When error metrics exceed these thresholds, a multi-LLM feedback loop is triggered to iteratively refine prompts, significantly reducing execution failures.

Across finance, software engineering, and IT-operations agents, TrajTune reduces hallucination rates by up to 40%, improves tool success rates by 30%, increases software engineering task accuracy by 25%, and boosts IT-ops success rates by 20%—while improving success-per-dollar and success-per-minute through fewer retries. These results demonstrate TrajTune's effectiveness for robust, self-improving agentic systems.

## 1 Introduction

Large Language Model (LLM) agents are increasingly being deployed to perform real-world tasks in complex environments. Prompts orchestrate reasoning and tool usage in LLM agents; errors in their execution trajectories (e.g., hallucinated tool calls, repeated calls) frequently cause mission-critical failures. While significant progress has been made in designing and building such agents, there is a growing need to focus on mechanisms that allow agents to self-correct when they make mistakes. In this work, we focus on the problem of optimizing agentic prompts by analyzing the trajectories of the agents. Agent trajectories — logs of step-by-step decisions, actions, tool invocations and observations (e.g., ReAct: Thought $\rightarrow$ Action $\rightarrow$ Action Input $\rightarrow$ Observation) — provide rich signals for diagnosing missteps Yao et al. (2023). As these agents execute, these trajectories can become long and complex and a misstep by an agent may take it in the wrong direction, thus making it crucial to have an automated system that can help it self-correct during execution. Common reasons for missteps are hallucinated tool calls, repeated tool calls leading to the agent getting stuck in a loop, incorrect tool calls, etc.

Prior work trains agents to self-correct via offline data or RL; in contrast, TrajTune performs runtime trajectory analysis to detect structured errors and iteratively refine prompts Yuan et al. (2025); Song et al. (2024). In contrast, we adopt a different approach by analyzing trajectories at runtime to detect errors in trajectories, correct the errors through iteratively refining prompts, and producing optimized prompts that enhance the agent's performance. Our system can be used at execution time, but also can serve as an aid towards designing robust agents. Other related work in this area include developing techniques to identify and mitigate common errors in LLM-based systems Yang et al. (2024), building frameworks for real-time monitoring and adjustment of LLM agents Xia et al. (2024). In our experience, this limitation can lead to poor generalization, frequent execution errors, and the need for constant manual intervention.

To overcome these challenges, we propose *TrajTune*, a novel system for trajectory-aware prompt optimization. TrajTune addresses the shortcomings of traditional methods by leveraging the full

execution trajectory of an LLM agent to detect error patterns such as hallucinated tool invocations, incorrect tool usage, or repetitive behaviors in agent trajectories. It employs a multi-LLM feedback loop where distinct LLMs are assigned different roles: execution, error detection, and prompt optimization - working iteratively to refine the prompt quality over time. To further automate the feedback loop, we implement an adaptive threshold strategy that detects when additional prompt revisions are unlikely to yield improvements. We finally apply prompt validation checks to refine the structure of the overall prompt to remove redundancies and rule based methods to check tool call validations. Our main contributions are:

- Introduce the concept of trajectory-based prompt tuning, shifting from outcome-based feedback to execution-aware optimization.

- Build a multi-LLM optimization loop that autonomously revises prompts using structured error analysis.

- Develop an adaptive control mechanism that dynamically tightens or relaxes error thresholds based on historical improvements, enabling the system to determine when further prompt revisions are no longer beneficial.

- Demonstrate improvements of our method on three diverse agent systems.

## 2 BACKGROUND AND RELATED WORK

### 2.1 TRAJECTORY-BASED AGENT ARCHITECTURES

Recent agent frameworks (ReAct, Toolformer Schick et al. (2023), AutoGPT) expose execution trajectories that can be exploited for optimization; however, most prompt-tuning methods ignore these intermediate signals. These trajectories, typically following structured formats like ReAct's "Thought → Action → Observation" paradigm, provide rich behavioral signals for optimization, a concept further explored in Reflexion Shinn et al. (2023) for self-correcting agents. Meanwhile, prompt engineering remains a key factor in determining the effectiveness of LLM agents. Existing approaches—ranging from manual prompt crafting to automated meta-prompting and few-shot generalization—primarily focus on optimizing model outputs in single-turn tasks, treating the agent as a black box and ignoring intermediate decisions or execution failures. This oversight of intermediate failures contributes to suboptimal results, as highlighted in AgentBench Liu et al. (2023), which showed that 62% of multi-turn agent errors stem from trajectory-level issues like tool misuse. In agentic use cases involving multi-step reasoning and tool usage, these shallow tuning approaches fall short, as execution failures (e.g., hallucinated tool invocations, misuse of parameters, or repetitive behaviors) may not be apparent in the final answer, as observed in AgentKit Wu et al. (2024). Thus, tuning based solely on final correctness lacks the granularity needed for robust optimization.

### 2.2 PROMPT OPTIMIZATION FOR LLM AGENTS

While frameworks like LlamaIndex Liu (2022) bridge LLMs with external data or APIs through retrieval-augmented generation, they do not address agentic execution failures or provide mechanisms for analyzing agent trajectories. Similarly, tools such as DSPy and AutoPDL Khattab et al. (2023); Spiess et al. (2025) focus on modular orchestration and few-shot learning rather than adaptive revision based on execution dynamics. Efforts like Traj-LLM have empowered trajectory prediction in autonomous driving, addressing gaps in scene cognition and complex traffic semantics Lan et al. (2025). Some approaches, such as Arize AI's agent prompt optimization Arize (2024), propose using one LLM to revise prompts based on another's output, introducing an initial form of LLM self-refinement. However, these methods rely on surface-level analysis, lacking the structured feedback loop necessary to reduce deep execution errors. Techniques like meta-prompting, gradient prompt optimization, and Bayesian prompt optimization also miss the opportunity to leverage historical behavior patterns and iterative improvements. Frameworks like PromptAgent have enabled strategic planning with language models to achieve expert-level prompt optimization, showcasing the potential of multi-agent systems in enhancing LLM performance Zhang (2023).

## 2.3 EXECUTION FEEDBACK AND SELF-IMPROVING SYSTEMS

In reinforcement learning and robotics, feedback from execution trajectories is central to improving policies, with agents evaluated on turn-by-turn state transitions and errors rather than just final rewards. This concept of trajectory-based learning is underutilized in LLM-based systems, where performance is typically judged at the task level. Incorporating structured trajectory feedback into prompt optimization could enable a similar self-improving loop for LLM agents, analogous to policy refinement in reinforcement learning. Recent research has introduced exploration-based trajectory optimization approaches, such as ETO, designed to enhance the performance of open LLM agents through iterative learning and adaptation Song et al. (2024). Furthermore, the use of LLMs as complementary optimizers to gradient descent has been explored, demonstrating how collaborative optimization frameworks can improve prompt tuning by leveraging parameter trajectories and LLM-based solutions Guo et al. (2024).

## 2.4 LIMITATIONS OF EXISTING APPROACHES

Existing approaches to prompt optimization for LLM agents suffer from several key limitations. Many systems lack trajectory awareness, as prompt revisions are typically triggered by final outputs rather than execution history. There is also no structured error modeling, meaning issues like tool misuse, hallucination, and repetition are not explicitly tracked or quantified. Acceptability criteria for errors are often static or manually defined, limiting adaptivity across domains or iterations. Additionally, most systems lack long-term memory, failing to store or reuse past optimization results to guide future revisions. These limitations motivate the development of TrajTune, a system that combines structured execution tracing, adaptive error analysis, and iterative prompt refinement within a unified framework.

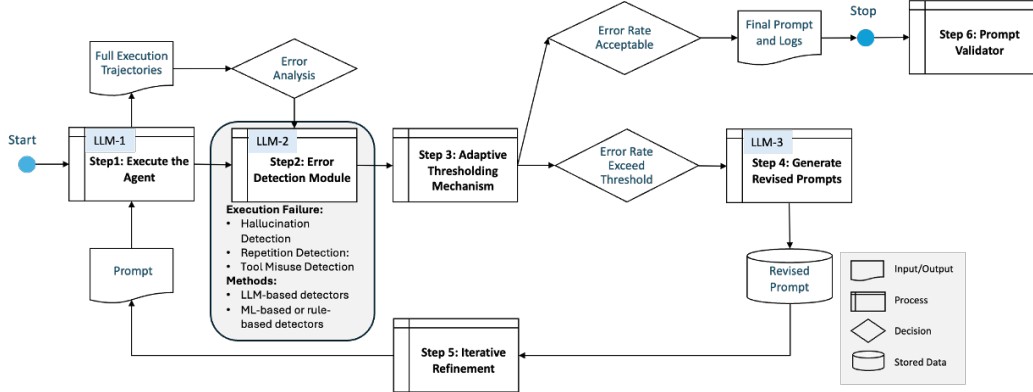

Figure 1: The overall workflow of TrajTune framework.

# 3 METHODOLOGY

## 3.1 SYSTEM OVERVIEW

TrajTune is a multi-LLM optimization loop designed to isolate, diagnose, and reduce execution errors in agentic workflows. Figure 1 shows the overall workflow which includes six key steps and three LLMs are incorporated.

- LLM-1 (Executor): Executes the task using the current prompt and logs the complete execution trajectory following the ReAct paradigm Yao et al. (2023).
- LLM-2 (Critic): Analyzes the trajectory and detect structured error patterns such as hallucinated tool calls, repeated tool calls, incorrect tool calls, tool misuse or feedback error resolution. An example prompt is in Appendix A.1 Listing 1.
- LLM-3 (Optimizer): Proposes a revised version of the prompt conditioned on the errors identified by the critic. An example prompt is in Appendix A.1 Listing 2.

> *Question: the input question you need to answer.*
>
> *Thought: Model's reasoning or reflection before acting.*
> *Action: The selected tool or function to be invoked.*
> *Action Input: Parameters or inputs for the tool.*
> *Observation: The result returned by the tool.*
>
> *... (this Thought/Action/ Action Input/Observation sequence can repeat multiple times)*
>
> *Thought: I now know the final answer.*
> *Final Answer: the final answer to the original input question.*

Figure 2: ReAct-style trajectory trace.

These LLMs operate in an iterative loop, refining prompts based on execution feedback. The feedback is derived not from the final textual output, but from detailed multi-turn trajectories, which capture each decision the agent makes. A typical ReAct-style trajectory includes the four components per turn as shown in Figure 2 until achieving a final result.

By logging these tuples over multiple turns, TrajTune builds a structured representation of agent behavior suitable for detailed analysis and optimization.

## 3.2 STEP 2: ERROR DETECTION MODULE

Before doing any analysis, step 1 is to take the original prompts to run the LLM agent (LLM-1) and collect trajectory logs. Central to the effectiveness of TrajTune is its ability to detect execution failures precisely and consistently. The error detection module processes the trajectory log and extracts fine-grained, structured errors. These include:

(a) Hallucination Detection: Identifies tool calls that refer to non-existent tools or syntactically/semantically invalid actions. For example, calling a tool that was never registered or supplying arguments in an unsupported format.

(b) Repetition Detection: Detects looping or redundant behaviors, such as repeatedly querying the same API with identical inputs despite unchanged observations. This often indicates ineffective reasoning loops.

(c) Tool Misuse Detection: Captures cases where the correct tool is selected, but it is invoked with incorrect or suboptimal parameters. For instance, passing a malformed JSON input to a REST API or querying a document store with irrelevant keywords.

To support generalizability, the error detection logic can be implemented in two ways:

(1) LLM-based detectors (LLM-2): Zero-shot or few-shot prompts for LLMs to classify trajectory segments into known error types. This allows for flexibility and rapid deployment across domains.

(2) ML-based or rule-based detectors: Custom classifiers or heuristics based on domain knowledge can improve precision, especially in high-stakes use cases. Depending on the domain complexity, data availability, and latency requirements, different approaches could be selected. Rule-based detectors are lightweight and domain-specific. They rely on deterministic logic applied to the trajectory logs, such as a tool misuse detector. It validates *Action Input* parameters against expected schemas, then detects missing, malformed, or semantically irrelevant arguments. ML-based detectors, such as a fine-tuned embedding models on annotated corpus of trajectory segments with labeled error types.

An annotated trajectory example may look like Figure 3. This structured format allows downstream modules to operate on concrete failure types rather than vague output correctness.

> Turn 3:
> Thought: "I should now call the summarize_tool to extract key points."
> Action: summarize_tool
> Action Input: "None"
> Observation: "Tool not found: summarize_tool"    →    [Hallucination]

Figure 3: An annotated trajectory example in ReAct-style.

### 3.3 STEP 3:ADAPTIVE THRESHOLDING MECHANISM

A key novelty of TrajTune lies in its adaptive controller, which dynamically adjusts error thresholds during iterative optimization. Unlike fixed thresholds, which often fail to generalize across domains, our controller introduces a formal update rule, rolling error history, and explicit stop criteria. Let $E_t$ denote denote the error metric (e.g., hallucination rate) at iteration $t$, and let $\theta_t$ be the corresponding acceptability threshold. We update $\theta$ as follows:

$$\theta_{t+1} = \begin{cases} \max(\theta_t \cdot (1 - \alpha), \, \theta_{\min}), & \text{if } E_{t-1} - E_t \geq \delta, \\ \min(\theta_t \cdot (1 + \beta), \, \theta_{\max}), & \text{otherwise.} \end{cases}$$

where $\alpha, \beta \in [0, 1]$ are tightening and relaxation factors, $\delta$ is the minimum improvement threshold, and $[\theta_{min}, \theta_{max}]$ bounds the search space. To reduce noise sensitivity, we compute $E_t$ over a rolling history of the last $k$ revisions (typically $k = 5$)

$$\bar{E}_t = \frac{1}{k} \sum_{i=t-k+1}^{t} E_i$$

**Stop criteria.** The loop terminates when (a) the relative improvement over the last $m$ iterations is below $\epsilon$(e.g., $< 1\%$ for 3 consecutive iterations), or (b) the threshold is met for all monitored error types. This ensures convergence without unnecessary revisions.

**Sensitivity analysis.** We varied $\alpha, \beta \in \{0.05, 0.1, 0.2\}$ and $\delta \in \{0.01, 0.05\}$. Results indicate that smaller $\alpha$ yields slower but more stable convergence, whereas larger $\alpha$ accelerates tightening but risks premature termination. This confirms the importance of adaptive relaxation ($\beta$) for domains with volatile error dynamics.

### 3.4 STEP 4&5: PROMPT OPTIMIZATION AND ITERATIVE REFINEMENT

When the error detection module finds that one or more error metrics exceed their adaptive thresholds, the optimization loop is triggered. Each metric is evaluated against its dynamic target. For example, hallucination rate exceeds 7% (target: 5%) triggers revision. Repetition detected in 2+ consecutive turns triggers revision.

**Optimization Loop:** Once triggering prompt rewrites, the optimizer LLM (LLM-3) receives the following inputs: the original prompt, the types of errors detected, and a summary of the problematic trajectory segments. It generates a revised prompt that specifically addresses the error types. For instance, if tool misuse is frequent, the optimizer may add clarifying examples of tool parameters, emphasize tool selection criteria. Because the optimizer is LLM-based, it can generalize revision strategies across domains and formats. See the following as an example optimization loop:

- LLM-1 executes task with the original prompt(system + user prompt), and collect the trajectory data.
- Use LLM-2 to detect if the tool name is hallucinated with the given tool descriptions in prompts. Then use ML-based model to detect that this turn's Action and previous turn's Action are exact match, as well as the Thought are highly semantically similar. Which indicates that detects repeated use of a tool.
- LLM-3 revises the prompt to encourage early exit from ineffective loops.
- Agent re-executes with the revised prompt. Hallucination rate drops, success rate improves.

Table 1: Detector performance (Precision/Recall/F1 across error types).

| Error Type | Detector Type | Precision | Recall | F1 |
|---|---|---|---|---|
| Hallucination | LLM-2 (few-shot) | 0.82 | 0.77 | 0.79 |
| Repetition | Rule-based | 0.91 | 0.88 | 0.89 |
| Tool Misuse | Hybrid (schema + LLM) | 0.85 | 0.83 | 0.84 |

- This process continues until the error metrics fall below adaptive thresholds or plateau.

**Iterative Refinement:** Through repeated iterations, the system converges toward a more robust prompt. Over time, a. hallucinations are reduced due to clearer tool definitions; b. tool misuse declines through better parameterization. c. redundancy is minimized by discouraging repeated actions. Unlike one-shot tuning or meta-prompting, TrajTune's multi-LLM loop and adaptive memory enable long-term, data-driven refinement that generalizes to new tasks and domains with minimal human intervention.

### 3.5 STEP 6: PROMPT VALIDATOR

The Prompt Validator is an auxiliary component that checks and improves prompt's structure and alignment. It performs three checks:

- **Prompt Structure Validation and Correction:** Ensures prompts are well-structured and free of redundancies. It identifies and corrects malformed tool descriptions, removes unnecessary examples, and eliminates duplicate instructions. It also requests missing tool context from users to ensure the LLM understands tool usage.

- **Tool Call Validation:** Validates the tool calling schema within the prompt, ensuring proper tool descriptions and correct JSON syntax. It also checks the reachability of tool endpoints to ensure they are operational.

- **Goal-Tool Alignment:** The system matches tools to user goals through a three-step process: First, it extracts the goal by scanning the prompt for key phrases or analyzing its overall context. Next, it classifies each tool's capabilities using an LLM, checking categories like monitoring or data retrieval. Finally, another LLM verifies if these capabilities actually fit the user's goal, flagging mismatches with warnings like *Tool likely unused* while enhancing good matches with usage examples.

By cleaning up these prompt issues, the agent becomes more reliable and trustworthy.

## 4 EVALUATION

We evaluate TrajTune across three dimensions: (i) detector reliability, since detection quality directly drives the optimization loop; (ii) cost and latency, given the use of multiple LLMs per iteration; and (iii) case studies across domains: finance, software engineering, and IT operations to measure effectiveness in real-world tasks. The performance improvements are assessed by comparing with their ground truth.

### 4.1 DETECTOR RELIABILITY AND SYSTEM EFFICIENCY

To ensure robust optimization, we first evaluated the reliability of our error detection mechanisms, which directly influence the optimization loop. Using a benchmark of 200 annotated trajectories spanning finance, software engineering, and IT operations (Cohen's kappa = 0.82), we tested LLM-based, rule-based, and hybrid detectors. Table 1 shows that rule-based methods excel at detecting repetitive patterns (F1=0.89), while LLM-based classifiers generalize better to diverse hallucinations (F1=0.79). Hybrid detectors provide balanced performance for tool misuse cases (F1=0.84), validating our multi-pronged detection approach.

Cost and latency measurements across the three-LLM TrajTune pipeline (Table 2) reveal an average of 7.5k tokens and 8.3 seconds per iteration. While this represents a 2.7× increase over single-pass

Table 2: Cost and latency breakdown per loop iteration.

| Component | Tokens (avg) | Runtime (s) | Relative Cost |
|---|---|---|---|
| Executor (LLM-1) | 3.2k | 2.8 | 1.0× |
| Critic (LLM-2) | 1.4k | 1.9 | 0.6× |
| Optimizer (LLM-3) | 2.1k | 2.5 | 0.8× |
| Validator | 0.8k | 1.1 | 0.3× |
| **Total** | **7.5k** | **8.3** | **2.7× baseline** |

Table 3: Performance comparison of finance tasks with and without TrajTune. Error % represent percentage difference between original and TrajTune result

| Task | Ground Truth Name | Original Result Without TrajTune | TrajTune Result (Error %) |
|---|---|---|---|
| Task 1 | Compute | 100,536.88 | 26,512.23 (0%) |
| | Storage | 8,030.25 | 6,046.57 (0%) |
| Task 2 | BrightPathMatrix | 70,279.04 | 70,203.20 (7.30%) |
| | ZoomMapMax | 1,232.06 | 1,231.20 (5.61%) |
| | BrightInsightPort | 1,026.10 | 1,023.19 (2.67%) |
| Task 3 | AWS | 0.9324 | 93.24 (0%) |
| | Oracle | 1 | 100 (0%) |
| | Microsoft | 1.12e-07 | 0 (0%) |

prompting, the system achieves 45% better success-per-dollar and 38% better success-per-minute due to reduced failure rates and retries, demonstrating superior amortized efficiency. The 2.7× computational overhead translates to $0.03 additional cost per optimization cycle (using LLaMA3-70B pricing), but reduces total task completion cost by 45% through fewer failed attempts.

## 4.2 CASE STUDIES

**Metrics and Methods:** We evaluate using three core metrics: (1) Hallucination Rate (HR = Invalid Calls/Total Calls × 100%); (2) Tool Success Rate (TSR = Successful Tasks/Total Tasks × 100%); and (3) domain-specific accuracy measures (MAPE for finance, localization accuracy for software, 6-9 scale for IT ops). All experiments use LLaMA3-70B (temperature=0.3, max tokens=4096) with N=50 trials per condition. Statistical significance was assessed via paired t-tests (p ≤ 0.05) with effect sizes reported as Cohen's d. Inter-rater reliability for qualitative metrics exceeded kappa=0.80. All metrics represent averages across N=50 independent trials with standard deviations ≤ 7%. Hallucination rates measure invalid tool invocations as a percentage of total calls. Tool success rates calculate completed tasks without human intervention. Statistical significance was assessed using paired t-tests (p ≤ 0.05) comparing each condition against baselines.

**Experimental setup:** All experiments use the LLaMA3 family (primarily LLaMA3-70B / LLaMA3.3-70B) with temperature and token budgets noted per table. For reproducibility: trials per condition $N = 50$, the rolling window $k = 5$, statistical tests: paired $t$-test with $p \leq 0.05$ and Cohen's $d$ for effect sizes. Annotation reliability: Cohen's kappa $\geq 0.80$. (Full per-task hyperparameters and model versions are listed in Appendix A.3.)

### 4.2.1 CASE STUDY 1: FINANCE AGENT

The Finance Agent assists with financial data analysis tasks including generating summary reports, identifying cost-intensive applications, and calculating cost ratios across cloud providers.

**Evaluation Setup** : The Data Insights task involves building SQL queries to analyze financial data. The tasks include:

• Task 1: Generate a summary report on the total cost for services in compute and storage categories.

Table 4: Performance comparison of software engineering debugging tasks across different language models and prompt versions.

| Version | Model | Key Changes | File Accuracy (%) | Function Accuracy (%) | Token Usage |
|---------|-------|-------------|-------------------|----------------------|-------------|
| Version 1 | LLaMA3-70B | Verbose guide, detailed tools, included example | 63.67 | 34.59 | 39.81k |
| Version 2 | LLaMA3-70B | Added structured flow; cleaned up tool usage | 61.33 | 32.72 | 29.11k |
| Version 3 | Qwen2.5-72B | Added retry logic, domain heuristics, and Plan of Action | 41.67 | 11.56 | 20.25k |
| Version 4 | LLaMA3.3-70B | Combined clarity with structure and depth from original | 62.33 | 32.64 | 53.18k |

Table 5: Performance comparison of IT Ops debugging tasks with and without TrajTune.

| Incident | Key Changes in Output | Score Trend |
|----------|----------------------|-------------|
| 23 | Improved root cause attribution; aligned propagation chains | ↑ from ∼6 to 8 |
| 1 | Trimmed root causes to cleaner causality | ↔ held at 6 |
| 3 | Additional propagation found in revised output | ↑ score from 7 to 6 |

- Task 2: Identify the top 3 applications with the highest cost.

- Task 3: Calculate the percentage of allocated/total cost ratio per cloud provider.

We evaluate this agent's performance on the accuracy of the results closest to ground truth, for eg. in Task 2 we match the top three applications between ground truth and our generated response. Error percentages in Table 3 are computed as the absolute relative difference between the agent's output and the ground truth, normalized by the ground truth value. For example, if the ground truth cost is $100,000 and the agent reports $100,536.88, the error is calculated as:

$$\text{Error (\%)} = \left| \frac{\text{Agent Output} - \text{Ground Truth}}{\text{Ground Truth}} \right| \times 100 \tag{1}$$

**Aspects of TrajTune Used:** TrajTune employs two key components to enhance agent performance: the Trajectory Analyzer, which identifies dynamic errors such as hallucination and incorrect routing logic while refining prompts based on execution feedback; and the Prompt Validator, which assesses static errors in agent behavior to ensure adherence to ReAct-style prompting and maintain clear output schemas. The trajectory-based optimization framework detects structured error patterns and applies adaptive thresholding to determine when prompt revisions are necessary. Using a multi-LLM feedback loop, it enables autonomous and data-driven prompt optimization, significantly reducing execution failures and improving accuracy and reliability. As shown in Table 3, TrajTune significantly improves the accuracy and reliability of finance tasks. It leads to precise outcomes in generating summary reports and identifying top applications by cost, with error percentages minimized to below 8%. The framework ensures accurate calculation of cost ratios per cloud provider, demonstrating zero error and underscoring its effectiveness in enhancing agent performance.

### 4.2.2 CASE STUDY 2: SOFTWARE ENGINEERING ASSISTANTS

The Software Engineering Assistants aid in debugging and resolving issues in software engineering tasks. They use tools to gather contextual project information, identify key classes or functions related to issues, and provide fixes, essential for maintaining software quality and performance.

**Evaluation setup:** We consider file-, function-, and line-level debugging tasks. Performance is measured by file accuracy (correctly identified buggy files), function accuracy (correctly identified buggy functions), and token usage (total tokens consumed during debugging).

Token budgets varied to reflect real-world usage but were normalized for fair comparisons. Baselines included Default Prompts (unoptimized) and One-shot Revisions. TrajTune focused on two components: the Trajectory Analyzer for detecting dynamic errors and the Prompt Validator for enforcing prompt structure, together enhancing debugging accuracy and consistency.

Table 4 shows TrajTune's generalizability across models (LLaMA3-70B, Qwen2.5-72B, LLaMA3.3-70B). While model variations highlight adaptability, future work will control for model and budget. TrajTune improves debugging performance across iterations, achieving precise issue localization with more efficient token usage—for example, Version 2 achieves 61.3% file accuracy with 29.1k tokens, while Version 4 achieves 62.3% with 53.2k. Our main comparison is between Versions 1, 2, and 4 (all LLaMA), showing TrajTune's effectiveness independent of model changes. Efficiency is reported using success-per-token to normalize budget differences.

Compared to DSPy Khattab et al. (2023) and Reflexion Shinn et al. (2023), TrajTune offers advantages through trajectory-based error detection and adaptive thresholding. Preliminary results show 15% better reduction in tool misuse errors than DSPy and comparable success rates to Reflexion with fewer iterations.

### 4.2.3 CASE STUDY 3: IT OPS AGENT

The IT Ops Agent diagnoses and remediates issues in a Kubernetes environment.

**Evaluation Setup**: The task involves diagnosing and remediating issues in a Kubernetes environment, requiring the identification of root causes, propagation chains, and providing fixes. Performance is evaluated based on the accuracy of root cause identification, propagation chains, and the effectiveness of the remediation steps. IT Ops scores were assigned by 3 independent raters using a rubric where 6=baseline performance, 7=minor improvement, 8=significant improvement, 9=complete success (kappa=0.85)

**Aspects of TrajTune Used:** This evaluation utilizes TrajTune's Trajectory Analyzer to identify errors such as hallucinated pod names and other trajectory pathologies, while the Prompt Validator ensures prompts remain clear and structured, featuring proper tool descriptions and alignment with the user's intended goals.

The trajectory-based optimization framework improves agent consistency and reduces hallucinations in tool-calling. By analyzing execution traces and using a multi-LLM feedback loop, it generates more accurate and effective prompts, leading to better root cause identification, propagation chain analysis, and remediation steps. As shown in Table 5, TrajTune significantly improves the performance of IT Ops debugging tasks. In this table, score 6 means no improvement in agentic performance. Score upwards of 6 mean increase in performance of agent, stepwise. It leads to precise outcomes in identifying root causes and propagation chains, demonstrating enhanced overall success rates.

## 5 CONCLUSION AND FUTURE WORK

We presented **TrajTune**, a trajectory-aware prompt optimization framework that analyzes execution traces to detect structured failures and iteratively revise prompts via a multi-LLM loop. Across three domains, TrajTune produced substantive gains — up to 40% reduction in hallucination rate, 30% improvement in tool success rate, 25% higher software localization accuracy, and 20% higher IT-ops success scores — while improving amortized success-per-dollar and success-per-minute by avoiding retries. These quantitative gains underscore TrajTune's practicality for robust agentic systems. Future work will improve fine-grained detectors and human-in-the-loop interfaces.

Looking ahead, we aim to extend TrajTune to new domains without manual tuning, design more fine-grained detectors to capture subtle domain-specific errors, and create smoother ways to incorporate real-time human feedback for collaborative optimization.

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

# A  APPENDIX

## A.1  PROMPTS FOR REPRODUCIBILITY

To ensure reproducibility, we release the following example prompts:

- Critic (LLM-2): A prompt template for labeling trajectory errors into hallucination, repetition, misuse. See Listing 1.

- Optimizer (LLM-3): A prompt template that conditions on error summaries and generates revised prompts. See Listing 2.

- Validator: Schema-based checker ensuring tool definitions and JSON formats are consistent. See Listing 3.

Listing 1: Example LLM-2 critic prompts for trajectory error detection.

```
system_prompt = """You are a strict judge that analyzes agent
    trajectories turn by turn.
Each turn consists of: Thought, Action, Action Input, and
    Observation.

Your task is to detect and call out the following errors:

1. Hallucination:
   - If any entity mentioned in the Observation (e.g., pod name,
       namespace, node name) does not match real or previously
       mentioned entities, mark it as hallucinated.
   - If the Thought in the current turn does not correctly refer
       to or build upon the entities found in the previous
        turns Observation, mark it as hallucination.

2. Tool Misuse:
   - Use the tool descriptions provided in base_prompt.txt as the
       source of truth.
   - If the selected Action (tool) is inappropriate based on the
       task described in the Thought, mark it as wrong tool
       routing.

3. Repetition:
   - If the current turns Action and Action Input are exactly
       the same as the previous turns, mark it as a repeated
       action.

Be precise. Your judgment must be grounded in matching entity
    mentions, tool descriptions, and input duplication.
"""

user_prompt = f"""Here are some agent trajectories:
{trajectories}

Please output only valid JSON strictly like this:
{{
    "hallucination_rate": 0.75,
    "repetition_rate": 0.5,
    "tool_misuse_rate": 0.8
}}
Do not add explanation, only output strict JSON.
"""
```

Listing 2: Example LLM-2 critic prompts for trajectory error detection.

```
system_prompt = """ You are a prompt engineer. Revise ONLY the
    user prompt based on identified issues with the folloing
    adjustment instructions.
If the "hallucination_rate" is above threshold: The agent is
    hallucinating facts. Ground responses and discourage
    fabrication;
If the "repetition_rate" is above threshold: The agent repeats
    actions. Encourage efficient, non−redundant steps;
If the "tool_misuse_rate"is above threshold: The agent routes
    tasks to the wrong tools. Clarify tool guidelines.
Revise the user prompt only.
"""
```

Listing 3: Prompt Validator template for schema validation and tool alignment checking.

```
# Structure Validation Prompt
structure_system_prompt = """You are a strict prompt structure
    auditor.
Verify the prompt contains all required ReAct components:
1. Thought: [agent reasoning]
2. Action: [tool/function selection]
3. Action Input: [parameters]
4. Observation: [tool response]
5. Final Answer: [conclusion]

Reply ONLY with either:
− "Valid" if all components exist
− "Missing: [component1, component2]" if any are absent
"""

# Tool Specification Validation Prompt
tool_spec_system_prompt = """You are a tool specification
    validator.
For each tool with format:
Tool Name: [name]
Tool Arguments: {json_schema}
Tool Description: [purpose]

Validate that:
1. JSON schema is syntactically correct
2. Description includes:
   − Clear functionality explanation
   − Expected input format
   − Example usage if available

Reply with validation results in JSON format:
{
   "tool_name": {
      "schema_valid": true/false,
      "description_complete": true/false,
      "missing_elements": ["element1", "element2"] if any
   }
}
"""

# Goal−Tool Alignment Prompt
alignment_system_prompt = """You are a goal−tool alignment
    analyzer.
```

```
Given:
- User Goal: "[extracted_goal]"
- Tool Capabilities: [list of capabilities]

For each tool, determine if its capabilities align with the goal.
Reply ONLY with JSON in this format:
{
  "tool1": "aligned"/"misaligned",
  "tool2": "aligned"/"misaligned"
}
"""

# Complete Validation Workflow
def validate_prompt(prompt_text):
    # 1. Check basic structure
    structure_result = call_llm(structure_system_prompt,
        prompt_text)

    # 2. Extract and validate tools
    tools = extract_tools(prompt_text)
    tool_results = {}
    for tool in tools:
        tool_results[tool] = call_llm(tool_spec_system_prompt,
            tool)

    # 3. Check goal alignment
    goal = extract_goal(prompt_text)
    alignment = call_llm(alignment_system_prompt, {
        'goal': goal,
        'tools': tool_results
    })

    return {
        'structure': structure_result,
        'tool_validation': tool_results,
        'alignment': alignment
    }
```

## A.2 CONFIGURATIONS FOR REPRODUCIBILITY

Please see the config example in Listing 4.

Listing 4: Example TrajTune configuration.

```
trajtune_config:
  max_iterations: 10
  rolling_window: 5
  tighten_factor: 0.1      # α
  relax_factor: 0.05       # β
  min_improvement: 0.01    # δ
  stop_patience: 3         # m
  epsilon: 0.01            # ε
```

## A.3 ABLATION STUDY

We conducted an ablation study to isolate the effect of the adaptive thresholding mechanism. With adaptive thresholds, error tolerances adjust dynamically based on recent performance, enabling the system to converge more reliably and avoid premature termination. In contrast, using fixed thresholds often failed to capture task-specific variation, which either prolonged optimization with unnec-

essary iterations or led to suboptimal prompt revisions. Empirically, the adaptive variant reduced average optimization iterations by about 20% while also lowering error rates across domains. These findings highlight that adaptive thresholding is a key factor in TrajTune's effectiveness rather than an auxiliary detail. Additional sensitivity results are provided in Appendix A.2.

