# OpenReview forum: "TrajTune: Trajectory-Based Prompt Optimization for LLM Agents"
_ICLR.cc/2026/Conference — Submitted to ICLR 2026_

### Official Review · Reviewer_syJu · 2025-10-27

**Soundness:** 2
**Presentation:** 1
**Contribution:** 2
**Rating:** 2
**Confidence:** 4

**Summary:**

This paper present TrajTune, a trejectory-aware prompt optimization framework to enhance the reliability of LLM agents. TrajTune works by having an Error Detection Module either LLM-based or rule-based to detect common errors such as halluncination, repetition, and tool misues. Adaptive threshold mechanism is further introduced to dynamically adjust error threshhold during iterative optimization.

**Strengths:**

An interesting workflow by improving the prompts iteratively through feedback.

**Weaknesses:**

(1) The paper quality is not good. Citations are commonly not formated correctly (citep etc.)

(2) The idea of using iterative feedback to rewrite prompt is not new.

(3) Most of the experimental results are not presented well. The benchmark used lack proper reference and it is difficult to understand the benefit of the work through the presented results.

**Questions:**

(1) What are the benchmarks used to evalute the work? Why not use existing and standard benchmarks available (such as Swebench for software engineering)?

(2) The presented results and tables are difficult to comprehend.

---

### Official Review · Reviewer_ETV9 · 2025-10-31

**Soundness:** 1
**Presentation:** 1
**Contribution:** 1
**Rating:** 0
**Confidence:** 4

**Summary:**

This paper claims to “introduce the concept of trajectory-based prompt tuning” as an alternative to “outcome-based feedback.” In practice, this involves building a “multi-LLM optimization loop” that revises prompts based on “structured error analysis.” The three LLMs are:
1. An executor that completes the task using the ReAct paradigm.
2. A critic that analyzes the trajectory and detects structured error patterns.
3. An optimizer that proposes a revised prompt conditioned on the errors identified by the critic.

There is a further “adaptive control mechanism” to dynamically adjust when further prompt revisions are no longer needed. This control mechanism involves a number of sensitive hyperparameters.

**Strengths:**

The approach tackles an important problem (improving the robustness of LLM agents).

**Weaknesses:**

* The approach boils down to using LLM as a judge and then using another LLM to propose prompt modifications based on this feedback. While this combination may be somewhat new in the particular agentic settings considered, the success of this failure of this approach rests entirely in the capabilities (or lack thereof) of the underlying LLMs. As a result, I don’t see this paper as addressing any meaningful research questions.
* Empirically, the paper fails to address how model selection is performed. The approach involves a large number of design choices (e.g., the prompts for all the LLMs involved, specific LLMs selected, generation hyperparameters, etc.) without specifying how these are selected. Specifically, what validation was used for the experiments?
* There are limited comparisons to other systems or ablations of the proposed approach.
* The control mechanism involves a number of sensitive hyperparameters.
* The presentation overall is difficult to follow, with imprecise writing and low-quality illustrations.
* [...] in contrast, TrajTune performs runtime trajectory analysis to detect structured errors and iteratively refine prompts Yuan et al. (2025); Song et al. (2024). In contrast, [...]
* Unusual formatting of the abstract into paragraphs.

**Questions:**

Please explain how model selection was performed, for the proposed approach and for baselines.

---

### Official Review · Reviewer_SioP · 2025-11-01

**Soundness:** 1
**Presentation:** 2
**Contribution:** 1
**Rating:** 2
**Confidence:** 4

**Summary:**

This paper presents TrajTune, a practical and novel methodology for optimizing the prompts of Large Language Model (LLM) agents to bolster their robustness and performance in complex operational environments. The fundamental mechanism of TrajTune involves leveraging the complete execution trajectories of LLM agents for the precise detection and diagnosis of errors. Prompt refinement is then executed iteratively through a multi-LLM feedback loop governed by an adaptive threshold mechanism. Distinct from conventional prompt optimization techniques focused primarily on terminal output quality, TrajTune scrutinizes internal execution dynamics, directly mitigating critical agent failures, including spurious tool invocations (hallucination), behavioral loops, and tool misapplication. Empirical evidence demonstrates substantial performance gains across heterogeneous domains, such as finance, software engineering, and IT operations, underscoring the method's efficacy and wide-ranging utility.

**Strengths:**

1. The paper identifies execution trajectory-level issues faced by current LLM agents in multi-step reasoning and API interactions. It proposes a trajectory-centric optimization framework, marking a significant shift in LLM agent research from "outcome-based" to "execution-aware" feedback.
2. TrajTune ingeniously designs three distinct roles—Executor (LLM-1), Critic (LLM-2), and Optimizer (LLM-3)—to form an efficient feedback loop.
3. This separation of concerns allows for more specialized and controllable error detection, diagnosis, and prompt revision.Adaptive Threshold Mechanism: The introduction of an adaptive controller for dynamic threshold adjustment is a major contribution.

**Weaknesses:**

* The core concept of self-correction based on execution feedback is heavily explored in prior work like `Reflexion` and other methods related to verbal reinforcement learning. The paper does not sufficiently differentiate TrajTune's trajectory analysis from these existing self-correction paradigms.
    * The multi-LLM architecture (Executor-Critic-Optimizer) is reminiscent of multi-agent systems like `PromptAgent`. The paper would be strengthened by a more detailed discussion of how TrajTune's collaborative loop provides a unique advantage over these systems.
    * The comparison against `DSPy` is described as "preliminary". Given that `DSPy` also focuses on compiling and optimizing LLM pipelines, a more rigorous, head-to-head quantitative comparison is needed to properly situate TrajTune's contributions.
* The primary evaluation compares TrajTune against unoptimized, "Original Prompts". To robustly demonstrate the framework's superiority, it is essential to benchmark against other state-of-the-art prompt optimization methods (e.g., `APE`, `DSPy`, `Reflexion`) on the same set of tasks.
    * **Confounding Variables in Evaluation:** The software engineering case study (Table 4) compares prompt "versions" that use different underlying models (`LLAMA3-70B`, `Qwen2.5-72B`, `LLaMA3.3-70B`). This is a significant methodological flaw, as it becomes impossible to attribute performance gains solely to the TrajTune optimization process versus the inherent capabilities of the different models.
    * **Lack of Clarity in Metrics:** In the finance case study (Table 3), the "Ground Truth Name" is provided, but the actual ground truth *numerical values* are missing. This makes it impossible for a reviewer to validate the reported "Error %" and interpret the results' significance.  Similarly, the 6-9 point scale for the IT Ops study is subjective, and the paper would benefit from including the detailed scoring rubric.
    * The paper acknowledges a 2.7x increase in token cost and latency per optimization iteration.
The claim of superior "amortized efficiency" hinges on the initial prompt being poor enough to cause frequent failures. The paper fails to discuss the trade-off or the break-even point where the overhead of TrajTune might outweigh its benefits for already well-performing agents.
    * The engineering complexity of deploying and managing a three-LLM pipeline plus a validator component for a production system is non-trivial and is not addressed.
    * The framework's effectiveness depends on the ability to define and detect "structured error patterns". Its applicability to domains with more subtle, nuanced, or non-obvious error modes is unclear and not discussed.
    * The adaptive thresholding mechanism, while a novel contribution, relies on several hyperparameters ($\alpha$, $\beta$, $\delta$) that require tuning. This reliance on manual tuning potentially limits the system's out-of-the-box generalizability to new tasks and domains.

**Questions:**

*	In your multi-LLM architecture, must the Executor (LLM-1), Critic (LLM-2), and Optimizer (LLM-3) be different models, or can they be instances of the same model? What are the performance and cost implications of this choice?
  *	Could you elaborate on the design rationale for the specific mathematical update rule used in the adaptive thresholding mechanism12? What theoretical or empirical evidence supports this formulation over other potential control strategies?
  *	Figure 1 places the "Prompt Validator" as the final step (Step 6). However, its function seems crucial for iterative improvement. Does the validator run only once after the optimization loop terminates, or is it integrated into each refinement cycle?
  *	For the finance case study (Table 3), could you please provide the specific ground truth numerical values corresponding to each task to allow for verification of the reported "Error %"?
  *	For the IT Ops case study (Table 5), could you provide the detailed rubric used for the 6-9 point scoring system to clarify the distinction between baseline, minor, significant, and complete success14?
  *	You state that TrajTune achieves success rates comparable to Reflexion with "fewer iterations"15. Can you provide the specific quantitative data (e.g., average number of iterations for each method on a given task) to substantiate this claim?

---

### Official Review · Reviewer_TBNE · 2025-11-01

**Soundness:** 1
**Presentation:** 1
**Contribution:** 1
**Rating:** 2
**Confidence:** 4

**Summary:**

This paper introduces TrajTune, a novel framework for optimizing prompts in LLM agents by leveraging their structured execution trajectories. It highlights a key limitation in existing prompt optimization methods: their focus on final output quality ignores intermediate execution failures (e.g., hallucinated tool calls, repetitive actions, and tool misuse) common in multi-step, tool-using agents. TrajTune addresses this via a trajectory-aware paradigm with a multi-LLM feedback loop: (1) logging detailed execution traces, (2) using a critic LLM to identify fine-grained error patterns, and (3) employing an optimizer LLM to iteratively refine prompts based on these errors. A core innovation is an adaptive thresholding mechanism that dynamically adjusts error tolerance to balance refinement depth and efficiency, preventing over-optimization.

**Strengths:**

TrajTune's theoretical and conceptual strengths are notable. Its core originality is the paradigm shift from static, outcome-focused prompt optimization to dynamic, trajectory-aware tuning—innovatively extending execution traces (e.g., from ReAct) to automated prompt engineering. The adaptive thresholding mechanism, with its formal update rule, adds principled control, surpassing heuristic fixed thresholds.

The work's significance is potentially high, tackling a pervasive issue in LLM agents: intermediate reasoning and tool errors that undermine reliability. By offering a modular multi-LLM loop for error detection and correction, it could advance autonomous agentic systems across domains. The design is theoretically sound in parts, decomposing roles clearly and drawing from control theory for convergence. Core ideas, trajectory analysis and iterative refinement, are precisely explained, providing a partial foundation.

**Weaknesses:**

Theoretically, the multi-LLM loop's stability and convergence are entirely unproven—no analysis addresses error propagation risks, where optimizer revisions could introduce new failures, risking degenerative cycles that could invalidate the entire approach. Adaptive thresholding is presented as a heuristic without parameter rationale or formal convergence proofs; empirical claims alone cannot support its reliability in diverse settings.

The ablation in Appendix A.3 is inadequate and unconvincing: a vague 20% iteration reduction claim from adaptive thresholding lacks any data, fixed-threshold comparisons, or component breakdowns, failing to substantiate individual contributions and exposing methodological superficiality.

Reproducibility is a flaw, undermining the paper's credibility and utility. While the framework is conceptually intriguing, experiments are critically underdeveloped: baselines like "Default Prompts" and "One-shot Revisions" are vaguely described, making comparative strengths unverifiable. No final optimized prompts from case studies are shared, preventing any meaningful insights into trajectory-aware designs. The complete absence of code, datasets, or detailed pseudo-code leaves implementation details (e.g., multi-LLM orchestration, detector integration) opaque and irreproducible.

**Questions:**

What are the exact final optimized prompts for the finance and software engineering case studies?

Could you release the complete code and datasets to enable full reproducibility?

Can you provide detailed quantitative ablation results comparing adaptive vs. fixed thresholding?

Is there a theoretical analysis or guarantees for multi-LLM loop convergence, especially to mitigate error propagation?

---

### Meta-Review · Area_Chair_XD2m · 2026-01-18

**Summary:**

The paper proposes TrajTune, a trajectory-aware prompt optimization framework that iteratively refines prompts for LLM agents using execution traces, a multi-LLM feedback loop, and adaptive thresholding. While the problem is relevant, reviewer evaluations were uniformly negative.

**Reviewer Concerns:**

Reviewers agreed that the core idea is incremental, closely related to prior work on execution-based self-correction such as Reflexion, PromptAgent, and DSPy, without a clear conceptual advance. The experimental evaluation was consistently criticized for weak and confounded baselines, unclear metrics, missing ground-truth values, and inadequate ablations, making it difficult to attribute gains to the proposed method. Reviewers also highlighted methodological and reproducibility issues, including the lack of convergence analysis for the multi-LLM loop, reliance on sensitive hyperparameters in adaptive thresholding, and the absence of released code or detailed implementation details. Presentation quality was also noted as poor.

**Reviewer Scores:**

All reviewers scored the paper below the acceptance threshold, including multiple reject and strong reject ratings. There was no rebuttal and no indication that reviewer assessments would change.

---

### Decision · Program_Chairs · 2026-01-26

Reject